# Progress and Impact of Latin American Natural Product Databases

**DOI:** 10.3390/biom12091202

**Published:** 2022-08-30

**Authors:** Alejandro Gómez-García, José L. Medina-Franco

**Affiliations:** DIFACQUIM Research Group, Department of Pharmacy, School of Chemistry, Universidad Nacional Autónoma de México, Avenida Universidad 3000, Mexico City 04510, Mexico

**Keywords:** chemoinformatics, compound databases, chemical space, diversity, drug discovery, open science, pseudo-natural product

## Abstract

Natural products (NPs) are a rich source of structurally novel molecules, and the chemical space they encompass is far from being fully explored. Over history, NPs have represented a significant source of bioactive molecules and have served as a source of inspiration for developing many drugs on the market. On the other hand, computer-aided drug design (CADD) has contributed to drug discovery research, mitigating costs and time. In this sense, compound databases represent a fundamental element of CADD. This work reviews the progress toward developing compound databases of natural origin, and it surveys computational methods, emphasizing chemoinformatic approaches to profile natural product databases. Furthermore, it reviews the present state of the art in developing Latin American NP databases and their practical applications to the drug discovery area.

## 1. Introduction

Natural products (NPs) are a major source of bioactive molecules, and their importance is invaluable [1]. Between 1981 and 2014, over 50% of newly developed drugs were developed from NPs [2]. Over nearly four decades, they have been a significant resource of bioactive compounds for medicinal chemistry [3]. There are several sources for bioactive molecules, which include marine [4,5], fungal [6,7], bacteria [8], and plants [9]. Endogenous substances produced by humans and animals are another vital source of bioactive compounds [10]. Venoms and poisons produced by different animals are other rich sources [11]. 

Currently, there is an effort to find bioactive compounds from NPs as starting points for the further development of drug candidates for infectious diseases: antibacterial [12], antiprotozoal [13], antifungal [14], and antiviral [15]. Additionally, NPs are currently employed in medicinal chemistry to develop new chemotherapies, for example, neurodegenerative [16], cancer [17], immune-related [18], liver [19], and kidney [20] diseases, to mention a few examples. Moreover, during the current pandemic outbreak, NPs have been a rich source for discovering potential lead compounds against severe acute respiratory syndrome coronavirus 2 (SARS-CoV-2) [21,22].

Figure 1 shows the chemical structures of representative NPs approved for clinical use. The figure shows the pharmacological effect and the source of the compound. With the exception of captopril, all compounds were approved for clinical use without modifying the original chemical structure of the compound found in the extraction source. Captopril was developed based on the bradykinin potentiating factor in *Bothrops jararaca* snake venom. In 1981, it was the first animal toxin-based drug approved for human use. [23,24]. Digoxin is obtained from the plants of the genus *Digitalis* [25].

Information regarding the known activities of plants, either of their therapeutic or side or toxic effects, can serve as a starting point in the drug discovery process from NPs [10]. Furthermore, the stress-driven growth of plants and micro-organisms is used in the drug discovery process from NPs since it stimulates the production of secondary metabolites [26]. On the other hand, NP-based drug repositioning is a technique with potentially lower development costs and shorter time frames [27]. NPs show great promise in drug repositioning because they have been used for various medical purposes for thousands of years [27].

Computer-aided drug design (CADD) [28] has helped to mitigate the cost of billions and decrease time through the preclinical and clinical phases [29]. Chemoinformatics is a discipline with many tools used in CADD that has deeply impacted drug discovery in the pharmaceutical industry and academia [30]. One definition of chemoinformatics is *the application of informatics methods to solve chemical problems* [31]. To date, the discovery process of more than 70 commercialized drugs has included a computational method [28]. Nowadays, chemoinformatics has major applications in the research of NPs to identify and optimize bioactive compounds [32,33]. In this context, databases of NPs play a key role in drug discovery. Over 120 different NP databases and collections have been published and re-used since 2000: 98 of them are still somehow accessible, and only 50 are open access [34]. Around the world, several NP databases have been published, which contain compounds found in a certain country or geographical region. Specifically, in Latin America, some databases have been published representing the biodiversity of a particular geographical area [35].

The present manuscript discusses the importance of NPs as a source of bioactive molecules, the relevance of compound databases in drug discovery research, and the role of chemoinformatics in the development and analysis of compound databases. Finally, it reviews the state of the art in developing Latin American NP databases and their practical applications in drug discovery.

## 2. Importance of Natural Products as a Source of Bioactive Molecules

Nature is an abundant source of privileged scaffolds. The term privileged structure was first proposed in 1988 as structures capable of providing useful ligands for more than one receptor [36]. However, in drug discovery, a privileged scaffold should not hit many targets as in the term “frequent hitters” because they are associated with unwanted effects [37,38]. Privileged scaffolds are sources of molecular skeletons around which one may build compound libraries in the search for new drug candidates [3]. Terpenoid, polyketide, phenylpropanoid, and alkaloid structures are examples of privileged scaffolds from NPs that are currently used in the design and development of new drug candidates (Figure 2) [39]. 

There is one approach that involves the preparation of biologically relevant small-molecule libraries through unprecedented combinations of NP fragments to afford novel scaffolds that do not occur in nature; these molecules are called “pseudo-natural products” (pseudo-NP). Pseudo-NPs retain the biological relevance of NPs yet exhibit structures and bioactivities not accessible to nature or through the use of existing design strategies. Pseudo-NPs may display unexpected bioactivities that differ from the activities of the NPs from which their fragments are derived. That is why their bioactivity should be monitored in a wide biological space through different biochemical and biological assays. Most of the pseudo-NP collections fall within the “Lipinski rule of 5” (Ro5) space, showing advantageous physicochemical “drug-like” properties. For the design of pseudo-NP libraries, it is important to consider that the combination of biosynthetically unrelated NP fragments may be beneficial for novel bioactivity, maximizing the biological relevance of the resulting pseudo-NP scaffold. There are pseudo-NP collections that have been developed through the first-time combination of some scaffolds, resulting in totally new chemical entities, such as chromopynones, indotropanes, pyrrotropanes, and pyrroquinolinones (Figure 3) [40,41].

Throughout history, NPs have served as biomolecule reservoirs, both for molecules that later ended up converting into approved drugs without suffering chemical modifications (Figure 1) and for starting points for optimization that later, with further structural modifications, were approved for clinical use. Sometimes, bioactive molecules from NPs lack suitable physicochemical properties, and their synthetic complexity may hinder their direct use as therapeutics. In this case, to be developed as drug candidates, NPs need to go through an optimization process that usually involves structural modifications to improve one or more of the following characteristics: potency, selectivity, solubility, metabolic and chemical stability, and the removal of toxicity (or at least a significant reduction in toxicity) [42]. This is usually done by decreasing the molecular size, eliminating the unnecessary functional groups and chiral centers, and introducing nitrogen atoms (because of the limited nitrogen presence in the NPs) [42].

## 3. Relevance of Compound Databases in Drug Discovery Research

CADD can potentially speed up and decrease the cost of the drug discovery process. Traditional drug discovery technologies have very low hit identification rates. For instance, the hit identification rate of high-throughput screening (HTS) is only 0.021% and of molecular docking is 34.8% [43]. Compound databases are very useful resources in CADD. A database can be defined as an organized collection of data in any field [44]. It is important to highlight the importance of databases, firstly as a starting point to organize information. Depending on the kind of information stored, databases can be divided into six categories summarized in Table 1 [45]. In order to retrieve the required information, it is important to identify and look into the correct database category. 

One major CADD approach for the identification of lead molecules is the virtual screening (VS) of compound databases [45]. The term VS was first mentioned in the 1990s [63], referring to the identification of novel hits from large chemical libraries. VS techniques are usually classified into two major categories: structure-based (SBVS) and ligand-based (LBVS). In general, SBVS is more suitable for finding structurally novel ligands and is the preferred method when the three-dimensional (3D) structure of the target protein has been experimentally characterized [64]. When the structure of the target is unknown, or its prediction by structure-based methods is challenging, LBVS is the choice [65]. LBVS assumes that molecules with similar structures exhibit similar behavior. Among the LBVS techniques are the quantitative structure–activity relationship (QSAR) [64] and quantitative structure–property relationship (QSRP) [66] studies. QSAR/QSPR studies aim to find a mathematical association between the molecule structure with a given property, such as biological activity [65]. In this sense, the bioactivity and chemical information (i.e., chemogenomic) databases are crucial to allow the creation of QSAR/QSPR models that predict certain pharmacological activity or a property of pharmaceutical interest for a determined molecule or set of analog molecules. 

Another important application of the databases in the drug discovery process is the training of artificial intelligence (AI) algorithms. AI encompasses a set of computational algorithms that allow computers to simulate human cognitive abilities such as learning from experience and solving problems [67]. Among the LBVS techniques is the AI-based QSAR, and the creation and training of these models rely on the data found in the bioactivity databases. AI can be applied to SBVS, specifically, to the docking of the protein-ligand complexes [68]. AI-based scoring functions have shown better performance in benchmark studies [69,70]. The creation of AI-based scoring functions depends on the availability of the required data in the database to train the model. AI algorithms have already been applied in the drug discovery process from NPs such as: data-mining into traditional medicines and peer-reviewed articles, the prediction of chemical structures from microbial genomes, the automation of the dereplication process of NPs, encoding NPs into molecular representations, the vectorization of NPs with molecular descriptors, the mapping of NPs in the chemical space, the engineering of likeness scores, and the deorphanization and generation de novo natural product-inspired compounds [71]. Finally, research on using AI to create models that allow the prediction of the biological effects of NPs has increased in recent years. The application of AI models to predict the biological effects of molecules, toxicity, and drug–target and drug–drug interactions has been reviewed elsewhere [72].

## 4. Role of Chemoinformatics in the Development and Analysis of Compound Databases

Generating a compound database relies on the capacity to represent chemical compounds so that the actual chemistry software can recognize and differentiate the molecules. For this purpose, several notations have been created that represent chemical structures. There are three types of notations for chemical structures: one-dimensional (1D), two-dimensional (2D), and three-dimensional (3D).

The most popular 1D notation is the simplified molecular input line entry system (SMILES), with its first version reported in 1998 [73]. A general issue with this notation is that the same molecule can be represented with multiple SMILES strings. Therefore, the canonical SMILES were developed: the canonicalization process allows the creation of unique SMILES strings for every molecule. It is important to be aware that multiple algorithms exist for canonicalization. Further, there is an extended version that allows stereochemistry specification: isomeric SMILES [74]. Most of the compound databases store the compounds using the SMILES notation. The international chemical identifier (InChI) [75] notation was first introduced in 2007 [76]. In contrast to SMILES, InChI allows the creation of a unique identifier for every molecule. Additionally, this notation allows the inclusion or exclusion of stereochemical, isotopic, and tautomeric information. Nevertheless, InChI was barely used: the reason could be that, in contrast to SMILES strings, it is not human-readable and has a long string. InChIKey strings appeared in 2009 to tackle the problems of InChI. It is a fixed-length (27-character) condensed version of InChI [76]. Later, SMILES arbitrary target specification (SMARTS) notation was developed to specify substructural patterns which allow the matching of molecules that contain the specified substructural pattern [77]. For 2D graphical representation, there are programs that allow drawing of the chemical structures and facilitate the storage and interconversion between standard 1D and 3D file formats [78]. 3D databases are very useful for structure-based screening. It is not common to find (high-quality) 3D databases, but among the resources that provide 3D high quality molecular representations is the ZINC database [58] which provides the protonated and tautomeric molecular form which is very important for molecular docking and other 3D-dependant applications [45].

Chemoinformatics has played a key role in database assembly, curation, and content analysis. Currently, there are available several open-source software that allow characterization of the physicochemical profile and structural features of compound databases. For instance, RDKit [79] is a collection of chemoinformatics and machine-learning software that is possible to use from Python or through a graphical interface with the free available software KNIME Analytics Platform [80]. RDKit allows the efficient calculation of several physicochemical properties of pharmaceutical interest from a large compound database. Examples are the octanol/water partition coefficient (logP) [81], topological polar surface area (TPSA) [82], molecular weight (MW), number of Lipinski hydrogen bond acceptors (HBA) and donors (HBD), and number of rotatable bonds (RB) [83,84]. Furthermore, with RDKit, it is possible to characterize the molecular complexity through the calculation of the number of stereocenters and the fraction of carbon atoms with sp3 hybridization. Additionally, this software allows users to identify and filter molecules with structural alerts: chemical moieties that can potentially confer toxicity to the molecule. There are more utilities of RDKit for the chemoinformatic analysis, characterization, and creation of compound databases: identification of the Murcko scaffold [85], molecule fragmentation, calculation of multiple fingerprints, and the generation of canonical SMILES, InChI and InChIKey strings. Moreover, it is suitable for the preparation of compounds for molecular docking studies. RDKit software has been extensively used in academia, as shown in these recent examples [86,87,88,89,90].

In the last ten years, chemoinformatic methods to evaluate the diversity of compound databases have been developed and adopted in the drug discovery process. Molecular diversity can be evaluated using the six physicochemical properties of pharmaceutical interest previously mentioned: logP, TPSA, MW, HBA, HBD, and RB [84]. Molecular diversity captures information regarding the whole molecule and is straightforward to interpret. It can be evaluated using boxplots, histograms, and density plots. In order to have a complete evaluation of the diversity, fingerprints help to capture structural information that the physicochemical descriptors do not. Fingerprints capture structural features using the minimum unit of information in informatics: the bit. A string made of just bits, containing only one and zeros, can be created for every compound in the database. Two common molecular fingerprints employed to capture structural information are the Molecular ACCess System (MACCS) keys-166 bits [91] and Extended Connectivity Fingerprint (ECFP4) [92]. With either of both fingerprints, it is possible to make similarity comparisons, using the Tanimoto coefficient [93], among the compounds in the database and even make comparisons between several databases. In this sense, the cumulative distribution functions allow the comparison of structural diversity quantitatively among several databases. The diversity of a compound database also can be computed by taking into account just the core structure of the molecule: the scaffold. In this regard, there are three different ways to evaluate scaffold diversity: counts, cyclic system retrieval curves, and Shannon entropy (SE). Finally, global diversity can be assessed using consensus diversity plots (CDPs). In CDPs, it is possible to represent four measures of diversity: the most common are fingerprint-based, scaffold, whole molecular properties associated with drug-like characteristics, and size of the database. All the different ways to assess the diversity of a compound database previously mentioned have been extensively reviewed recently [94]. Additionally, the reader is further directed to the following references for more detailed information about the basis of molecular diversity analysis [95,96]. There is a free-access online server for diversity assessment that uses, as an input, the SMILES strings and allows the evaluation of diversity, creating the plots mentioned above in an automated way: box plots, histograms, and density plots from the logP, TPSA, MW, HBA, HBD and RB, cumulative distribution functions, cyclic system retrieval curves, CDPs, and SE determination [97].

## 5. Natural Product Databases

Between 2000 and 2019, 123 commercial and open access NP databases have been published. Of them, 98 are still somehow accessible, 92 are open access, and only 50 contain molecular structures that can be retrieved for a chemoinformatic analysis [34]. Table 2 summarizes examples of the most representative NP databases. Among the largest commercial databases is the Dictionary of Natural Products [98]. It contains more than 230,000 compounds and provides names and synonyms, physicochemical properties, spectroscopic data, molecular structures, and biological source and use. Another commercial database is Scifinder [99], assembled and maintained by the American Chemical Society (ACS). It contains arguably the most extensive collection of NPs, with over 300,000. This is due to the fact that, since 1957, the Chemical Abstracts Service (CAS), a division of the ACS, assigns a unique registry number to every new chemical substance reported in the scientific literature. Another large commercial database is Reaxys [100], collected and maintained by Elsevier. It contains approximately 10^7^ molecules including over 200,000 NPs. The Collection of Open Natural Products (COCONUT) [101] is a major open access database of NPs, containing more than 411,000 NPs collected from 50 open access NP databases. The Universal Natural Product Database [55] is a compilation that tried to gather all the known NPs; it has more than 229,000 NPs. It provides 3D structures with stereochemical information and calculated molecular descriptors. It is not yet accessible through the link in the original publication. Instead, it is contained and maintained on the ISDB website [102]. The SuperNatural Ⅱ [103] database contains over 325,000 NPs and includes information about 2D structures, physicochemical properties, predicted toxicity class, and potential vendors. Nevertheless, it does not provide a bulk download. 

ZINC [104] is another open access database with over 80,000 NPs, with approximately 48,000 which are purchasable. It includes information regarding known biological targets and predicted targets. The download of the entire subset of NPs in 1D or 3D notation is straightforward. Some NP databases are no longer accessible through the link provided in the original publication. Fortunately, their structures are in ZINC. Such is the case with the Herbal Ingredient Targets [105] and Herbal Ingredients in vivo Metabolism database [106], which contain NPs mostly from Chinese plants. Specs [107] has an industrial catalog of purchasable NPs, although the website does not allow the downloading of compounds anymore. Nonetheless, the structures are available via ZINC. Despite the Universal Natural Product Database, SuperNatural Ⅱ, and ZINC being among the largest databases of NPs in the public domain, they do not offer information regarding the taxonomic and geographic origin of the organisms that produce the NPs, and there is a lack of literature references [34].

Traditional Chinese medicine (TCM) is part of the public health system [108]. Therefore, the China Government encourages research in the area of NPs, and as a consequence, a large number of NP databases have been published [109,110,111,112,113,114,115]. Nonetheless, TCM@Taiwan is the most extensive database of NPs used in the TCM [116], containing approximately 58,000 molecules. Regarding traditional medicine in India (Indian Ayurveda), there are two open access databases available: IMPPAT [117], which contains more than 10,000 phytochemicals extracted from 1700 medicinal plants; and MedPServer [118], containing 1124 NPs coming from North-East India. Moreover, there are several databases containing compounds from African traditional medicine [119,120,121,122,123,124]. Nevertheless, AfroDB [125] is the most comprehensive, composed of around 1000 NPs, and it is accessible via ZINC.

**Table 2 biomolecules-12-01202-t002:** Most representative natural products databases.

Database Name	Number of Compounds	Accessibility	Reference
Collection of Open Natural Products (COCONUT)	411,621	Open access	[101]
Universal Natural Product Database	∼229,000	Open access	[55]
SuperNatural Ⅱ	325,508	Open access	[103]
ZINC	∼80,000	Open access	[104]
Dictionary of Natural Products	∼230,000	Commercial	[98]
Scifinder	∼300,000	Commercial	[99]
Reaxys	∼200,000	Commercial	[100]
TCM@Taiwan	∼58,000	Open access	[116]
IMPPAT	∼10,000	Open access	[117]
AfroDB	∼1000	Open access	[125]

## 6. Latin American Natural Product Databases

Around the world, several NP databases are published that represent the biodiversity of a specific geographical region. For instance, the databases mentioned in Section 5 represent the biodiversity of China, India, and Africa. Latin America stands out for its rich and unique biodiversity. In fact, it is home to at least a third of the global biodiversity [126]. Therefore, the Latin America region is a potential source of new drug candidates. Some Latin American countries have published their own NP database that contains compounds found in their respective country. Table 3 summarizes the Latin American NP databases released so far. In the next subsections, each database is discussed. 

### 6.1. NuBBE_DB_

The database is the result of the collaboration between the Nuclei of Bioassays, Biosynthesis and Ecophysiology of Natural Products (NuBBE) research group of the São Paulo State University and the Laboratory of Computational and Medicinal Chemistry of the University of São Paulo. The database was published in 2013 as the first NP library of Brazilian biodiversity, containing 640 compounds [127]; in 2017, an updated version came out with more than 2000 NPs [128]. Currently, the database contains 2223 compounds. The available information regarding the compounds includes the International Union of Pure and Applied Chemistry (IUPAC) name, linear notations (SMILES, InChI, and InChIKey strings), Ro5 and Veber descriptors, and predicted spectroscopic data: nuclear magnetic resonance (NMR), source, therapeutic effect and reference. It is possible to download the whole database in .mol2 format. Additionally, the database can be found in Chemspider and ZINC, and it is part of the COCONUT database. 

The website allows users to search compounds by selecting specific criteria: metabolic class (alkaloids, flavonoids, lignoids, etc.), name and location of the species that contain the NP, source (marine, plant, etc.), and drug-like physicochemical properties. Furthermore, one can draw a structure and retrieve the compounds that contain it or search compounds that contain a specific NMR signal.

An absorption, distribution, metabolism, excretion and toxicity (ADMET) profile of the database revealed that 91% of the compounds can permeate through the human intestinal barrier, and 93% of the molecules can efficiently move in systemic circulation and reach their desired site of action. Moreover, it is predicted that most of the compounds do not inhibit five isoforms of CYP450 (CYP 3A4, 2D6, 1A2, 2C9, and 2C19). The CYP450 enzyme is responsible for detoxifying more than 80% of drugs in liver first-pass metabolism, and therefore, any compound that inhibits it may cause toxicity. The clearance prediction revealed that 94% of the compounds are readily excreted from the human body after executing their therapeutic function. Finally, 87% of compounds were shown to have no mutagenicity, tumorigenicity, reproductive effect, and irritant properties [137].

Another study characterized the chemical space and diversity. It was found that NuBBE_DB_ has a focused chemical space within the space of drug-like physicochemical properties. The study also revealed that the larger source of diversity is driven by the side chains. Another finding revealed that the diversity and complexity varies according to the origin of the compounds when comparing NuBBE_DB_ to other NP databases. One conclusion of the study is that NuBBE_DB_ is a promising source of molecules for drug discovery [138].

The NuBBE_DB_ database was employed in a VS study with the purpose of finding compounds against Trypanosoma cruzi. The researchers looked for trypanothione reductase inhibitors: this enzyme is a validated target for the discovery of new antiprotozoal compounds. Ten compounds were identified as potential inhibitors of the enzyme [139]. In another study, 13 compounds against Mycobacterium tuberculosis were identified from NuBBE_DB_ [140]. The molecules are inhibitors of the serine/threonine protein kinase, which is essential for the growth and survival of the pathogen [141].

### 6.2. SistematX

The database was developed at the Laboratory of Cheminformatics of the Federal University of Paraiba, Brazil. The first version came out in 2018 containing 2150 secondary metabolites [129], and a second version was published in 2021 with a total of 9514 unique secondary metabolites [130]. The information for every compound includes the IUPAC name, SMILES, InChI and InChIKey strings, CAS registry number, physicochemical drug-like descriptors, predicted NMR spectra, predicted biological activities, and the bibliographic reference. A unique feature is the information regarding the taxonomic rank, from family to species, and the global positioning system (GPS) coordinates of the plant from which the compound was isolated. On the website (Table 3), the search of specific compounds can be through the 2D drawing of the structure, by the SMILES strings, compound name, taxonomic rank, and physicochemical properties. It is possible to download the entire database in .csv or .sdf format.

SistematX has been employed in five VS studies. In the first study, compounds with potential antichagasic activity were identified from 1306 sesquiterpene lactones on the database. (Chagas disease is an endemic disease caused by Trypanosoma cruzi.) The study employed two approaches, LBVS and SBVS. From LBVS, the most prominent compound showed a probability of 0.82 of inhibition. From SBVS, 13 potential inhibitors were identified [142]. In another VS study, with the purpose of identifying compounds against the intracellular parasitic protozoan Leishmania donovani which causes Leishmaniasis, 13 promising, enzyme-targeting, antileishmanial compounds were identified from the sesquiterpene lactones on SistematX [143]. In the third VS study, the researchers looked for compounds against Schistosoma mansoni, which causes the chronic parasitic disease Schistosomiasis. From the 1000 alkaloids on SistematX, five compounds were identified with potential multitarget schistosomicidal activity [144]. In the fourth VS study, 1955 diterpenes on SistematX were employed to search for compounds against SARS-CoV-2. Nineteen compounds were identified as potential SARS-CoV-2 inhibitors [145]. In the most recent VS campaign, the researchers were seeking acetylcholinesterase (AChE) inhibitors, which is an approach for the treatment of Alzheimer’s disease. They employed a combined approach in which machine learning classification models and molecular docking calculations were used to identify two promising AChE inhibitors [146]. Other applications of SistematX include chemotaxonomic studies using self-organizing map algorithms [147] and the profile of over 2000 metabolites from the Asteraceae family while screening for inhibitors of Leishmania major dihydroorotate dehydrogenase [148].

### 6.3. UEFS

The NP database of the State University of Feira de Santana [131] was developed and is maintained by the State University of Feira de Santana in Bahia, Brazil (UEFS, for its acronym in Portuguese: *Universidade Estadual de Feira de Santana*). The database contains NPs that have been separately published, but there is no common publication nor public database for it. Nevertheless, it is accessible via ZINC. There are 503 NPs in the database. It is possible to download the whole database in .mol2 or .sdf format, and it provides a bulk download of the SMILES strings. The available information of the NPs includes calculated physicochemical properties, biological targets, and binding affinity, together with the bibliographic reference. There is a cross-reference for the biological targets to Reactome which is an open source, open access, manually curated and peer-reviewed pathway database [149]. Finally, it is possible to find information about the vendors of individual compounds.

### 6.4. CIFPMA

The NP database of CIFLORPAN from the University of Panama, Republic of Panama (CIFPMA) was developed by the Center for Pharmacognostic Research on Panamanian Flora (CIFLORPAN, for its acronym in Spanish: *Centro de Investigaciones Farmacognósticas de la Flora Panameña*), College of Pharmacy of the University of Panama. The first version was published in 2017 [132], containing 354 molecules; in 2019, the database was updated to 454 compounds [133]. The compounds have been tested in over 25 in vitro and in vivo bioassays, for different therapeutic targets including anti-HIV (human immunodeficiency virus), antioxidants, and anticancer. In fact, the compound structures are available upon request.

A chemoinformatic analysis of the database suggested that, in general, the compounds have drug-like properties. The database was compared to the TCM@Taiwan and UEFS databases mentioned in Section 5 and Section 6.3 and other NP databases. It was found that CIFPMA has the largest scaffold diversity compared to other databases. Moreover, unique scaffolds were found in the CIFPMA database. Finally, it was established which scaffolds are present in compounds with experimental cytotoxic effect, anti-HIV-1, antimalarial, anti-trypanosomatid, and antifungal activities [132].

The database was part of another chemoinformatics study, which involved a comparison of several NP databases against other databases with compounds of synthetic origin. The study revealed that so many of the NPs and synthetic compounds share the same chemical space. Moreover, the NPs present a larger fingerprint-based diversity than the synthetic compounds. Furthermore, the study revealed that NPs have a higher proportion of chiral carbons and atoms with sp^3^ hybridization and greater complexity, while synthetic products contain a greater proportion of aromatic atoms. Lastly, cyclicity, relative shape, and flexibility are very similar in NPs and synthetic compounds [133].

### 6.5. UNIIQUIM

The database was created at the National Autonomous University of Mexico (UNAM, for its acronym in Spanish: Universidad Nacional Autónoma de México) by The Informatics Unit of the Institute of Chemistry (UNIIQUIM, for its acronym in Spanish: *Unidad de Informática del Instituto de Química*). The database [134] is composed of NPs from Mexico and mainly NPs isolated and characterized by the Department of Natural Products of the Institute of Chemistry, UNAM. The number of NPs on the database is not clear, and the website is only in Spanish. The information on the NPs includes the IUPAC name, CAS registry number, physicochemical properties, the species that synthesizes the NP, the spectroscopic techniques employed to characterize the compound, experimental biological activity, and reference to either the article where the NP is reported or to the articles that report the biological activities. In the current version, it is not possible to make a bulk download. The content can be browsed displaying a table either with the chemical structures or with the producing organism. Furthermore, the content can be browsed in a table that contains the bibliographic references.

In a study, the chemical and toxicological profile of molecules with analgesic activity was described. The results showed that most of the compounds probably interact with the opioid receptor. Moreover, the predicted acute toxicity is low, and none is predicted to be mutagenic. The study concludes that due to the structural diversity, the common nociception activity and the predicted safety profile as nonmutagenic agents highlights the importance of the molecules for further studies on the search of analgesic and nociception effects [150].

### 6.6. BIOFACQUIM

The database was curated and constructed in Mexico by the Computer-Aided Drug Design at the School of Chemistry (DIFACQUIM, for its acronym in Spanish: *Diseño de Fármacos Asistido por Computadora*) research group, UNAM. The first version came out in 2019 [135] and contained 423 NPs isolated and characterized in Mexico at the School of Chemistry, UNAM, between the years 2000 and 2018. Later, in 2020, a second version came out [136], and the database was updated with NPs isolated and characterized by research groups of other Mexican institutions, reaching a total of 531 molecules. Nowadays, the database contains 553 NPs. The database is composed mainly of NPs that come from plants, followed by fungus, and to a lesser extent, propolis and marine animals. There is a website for the first version of the database, and it allows the user to search the compounds by name. Moreover, it is possible to retrieve compounds by kingdom (plant, fungus, propolis). The entire database can be downloaded in .csv format. The latest version of the database is available on a different website [136], and it is possible to download the whole database in .sdf format. Information on the NPs includes the compound name, SMILES strings, bibliographic reference, taxonomic rank (kingdom, genus, species), place where it is found, the source from which the NP was isolated, biological activity, and *IC_50_* value. The database is also available at ZINC, and it is part of the COCONUT database.

A chemoinformatics analysis of the first version of the database concluded that the compounds have a broad coverage in the chemical space and overlap regions in the drug-like space. Furthermore, compounds very similar to drugs approved for clinical use were identified [135]. In another study, a structural content analysis of the second version was performed. BIOFACQUIM was compared to ChEMBL 25 (1,667,509 molecules) and a database with 169,839 NPs. The researchers concluded that 44.3% of the unique compounds contained in BIOFACQUIM are focused on drug-like space in terms of physicochemical properties. Additionally, a significant number of compounds and scaffolds (79 and 29, respectively) were identified that were not present in the two large reference sets [136]. Finally, an in silico absorption, distribution, metabolism, excretion and toxicological (ADMET) profile of the second version of BIOFACQUIM was performed. The study concluded that the absorption and distribution profiles of the compounds in BIOFACQUIM are similar to those of approved drugs, while the metabolism profile is comparable to that in other NP databases. The excretion profile of the compounds is different from that of the approved drugs, but their predicted toxicity profile is comparable [151].

An independent VS study looked for beta-glucosidase inhibitors. The pharmacological applications of these compounds include obesity, diabetes, hyperlipoproteinemia, cancer, HIV, and hepatitis B and C. Employing classification models (two-variable artificial network), eight compounds were identified from BIOFACQUIM as active [152]. In addition, in an independent study, Barrera-Vázquez et al. looked for senolytic compounds which selectively eliminate senescent cells. Cellular senescence is a cellular condition that involves significant changes in gene expression and the arrest of cell proliferation. The elimination of senescent cells delays, prevents, and improves multiple adverse outcomes related to age. Through the use of chemoinformatics tools (fingerprinting and network pharmacology), and employing two NP databases, InflamNat and BIOFACQUIM, three senolytic compounds were identified [153].

Table 4 summarizes the main applications of databases of representative Latin American natural products to identify bioactive compounds.

## 7. Conclusions and Perspectives

Nature is a significant source of structurally novel compounds that remains far from being fully explored. NP databases play an important role in the drug discovery process, serving as a systematic and organized source of potential novel hit and lead molecules. Several chemoinformatic methods have been used to organize, characterize, and mine different NP databases, identifying promising molecules. Nevertheless, many obstacles slow down the drug discovery from NPs driven by chemoinformatics approaches. Firstly, not all the NP databases are open source, restricting the access to a certain number of research groups with enough resources to pay for the access. Even if a research group has sufficient resources to pay for access, it will always be more attractive to resort to an open access database. As a consequence, myriads of NPs will remain inaccessible due to the payment restriction. On the other hand, access to many open access NP databases is not possible anymore; thus, invaluable information is lost, perhaps forever. The number of countries and research groups that curate and create NP databases is limited; just a few countries have tried to characterize NPs specific to their geographical region. Therefore, an incalculable number of novel molecules are still to be discovered. Nowadays, the number of open access and still available NP databases is limited. Therefore, there is a sense of urgency to keep curating and creating new NP databases.

Latin America stands out for its rich and unique biodiversity, which maybe encompasses a third of global biodiversity [126]. Regardless, just a few Latin American countries have gathered and characterized NPs from their region in a database. As far as we know, research groups in Colombia, Peru, and El Salvador are currently building compound databases to be released in the future. Previously, the need for a unified NP database that represents the biodiversity of Latin America has been pointed out [35]. Currently, in Mexico, the DIFACQUIM research group, in collaboration with several other countries in Latin America, is working on the creation and curation of a NP database that will gather all the NP databases of Latin America. The construction is in an early stage. Nevertheless, it will try to encompass the actual published NP databases and the upcoming ones. 

In this review, we also surveyed the practical applications of the Latin American NP databases in medicinal chemistry. It was concluded that most of the Latin American NP databases had been used as a basis to identify multiple promising candidates to be considered for further development for the treatment of numerous diseases. The growth of the practical applications of the Latin American NP databases is anticipated in the near future.

## Figures and Tables

**Figure 1 biomolecules-12-01202-f001:**
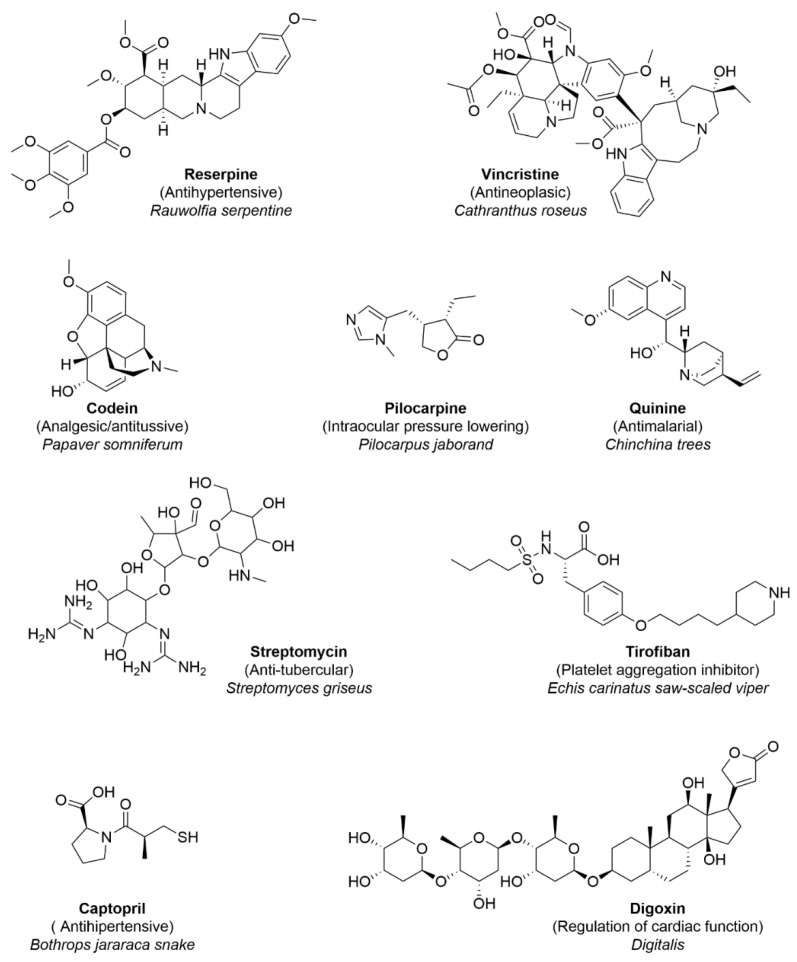
Chemical structures of representative natural products approved for clinical use. The pharmacological effect and the source of the compound are indicated (plants, animals, and bacteria). Captopril was inspired by a natural product (see main text for details).

**Figure 2 biomolecules-12-01202-f002:**
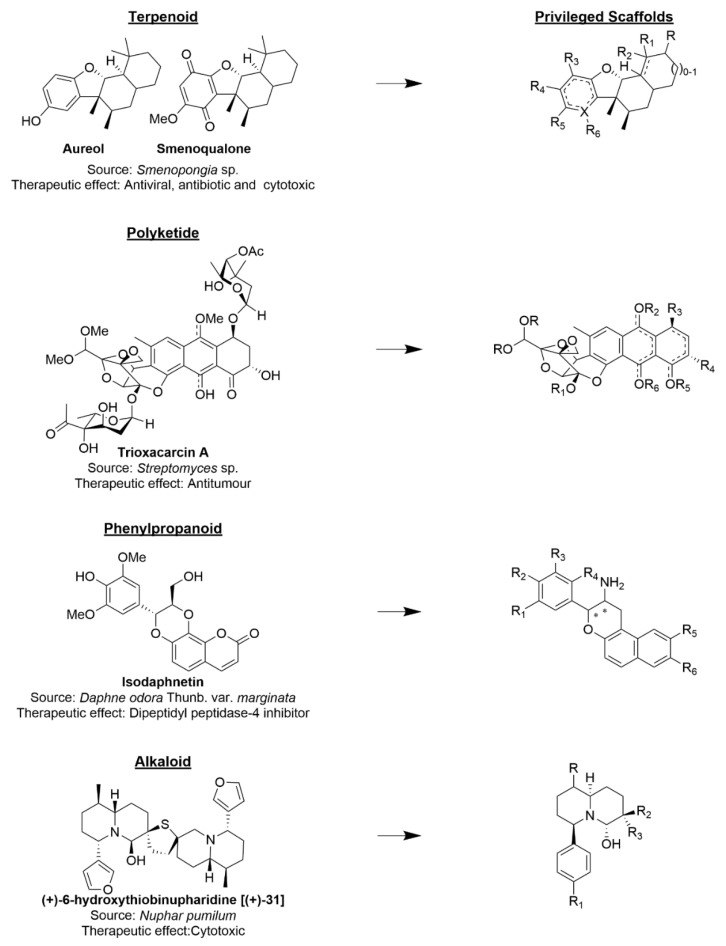
Examples of privileged scaffolds present in natural products.

**Figure 3 biomolecules-12-01202-f003:**
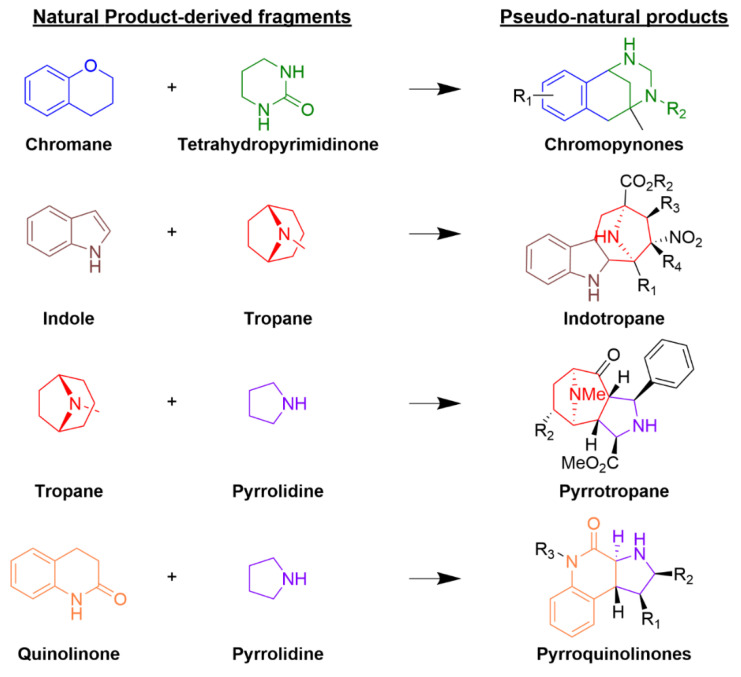
Examples of the combination of NP-derived fragments to form pseudo-NPs. The resulting pseudo-NPs come from a specific synthetic route that is depicted elsewhere [40].

**Table 1 biomolecules-12-01202-t001:** Categories into which databases can be divided according to the type of information stored.

Database Category	Content	Database	References
Chemical information	Chemical and crystal structures spectra Reactions and syntheses Thermophysical data	ChemSpider ChEBI Chemical Universe Database GDB	[46] [47] [48]
Bioactivity	Inhibitor constant (*K_i_*) Dissociation constant (*K_d_*) Half maximal inhibitory concentration (*IC_50_*) Half maximal effective concentration (*EC_50_*)	PubChem ChEMBL BindingDB ChemBank PDBbind	[49] [50] [51] [52] [53]
Drug	Detailed drug data Comprehensive drug target information	DrugBank	[54]
Natural product	Pathways (synthesis and degradation) Structures	Universal Natural Product Database MeFSAT Natural Product Atlas	[55] [56] [57]
Chemical availability	Available compounds offered by chemical vendors	ZINC NCI	[58] [59]
Fragment	Structures Physicochemical information Binding site preferences	FDB-17 Fragment Store PADFrag	[60] [61] [62]

**Table 3 biomolecules-12-01202-t003:** Latin American natural products databases.

Database	Size	Country	Source	Database Website	Reference
NuBBE_DB_	2223	Brazil	Plants Microorganisms Terrestrial animals Marine animals	http://nubbe.iq.unesp.br/portal/nubbe-search.html	[127,128]
SistematX	9514	Brazil	Plants	https://sistematx.ufpb.br/	[129,130]
UEFS	503	Brazil	Plants	http://zinc12.docking.org/catalogs/uefsnp	[131]
CIFPMA	454	Panama	Plants	Not available. Structures accessible under request.	[132,133]
UNIIQUIM	Unknown	Mexico	Plants	https://uniiquim.iquimica.unam.mx/	[134]
BIOFACQUIM	553	Mexico	Plants Fungus Propolis Marine animals	Database version 1 https://biofacquim.herokuapp.com/ Database version 2 https://figshare.com/articles/dataset/BIOFAQUIM_V2_sdf/11312702	[135,136]

**Table 4 biomolecules-12-01202-t004:** Practical applications of the databases of Latin American natural products.

Database Name	Disease or Symptom	Causative Agent	Number of Identified Compounds	Reference
NuBBE_DB_	Chagas disease	*Trypanosoma cruzi*	10	[139]
	Tuberculosis	*Mycobacterium tuberculosis*	13	[140]
SistematX	Chagas disease	*Trypanosoma cruzi*	13	[142]
	Leishmaniasis	*Leishmania donovani*	13	[143]
	Schistosomiasis	*Schistosoma mansoni*	5	[144]
	Coronavirus disease 2019	SARS-CoV-2	19	[145]
	Alzheimer’s disease		2	[146]
UNIIQUIM	Pain		6	[150]
BIOFACQUIM	Obesity		8	[152]
	Diabetes			
	Hyperlipoproteinemia			
	Cancer			
	HIV/AIDS *			
	Hepatitis B and C.			
	Age-related diseases		3	[153]

* Human immunodeficiency virus infection and acquired immunodeficiency syndrome (HIV/AIDS). Although CIFPMA does not appear in the table, their compounds have been assayed in a wide range of in vitro and in vivo bioassays.

## Data Availability

Not applicable.

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
