# Peer review of "Progress and Impact of Latin American Natural Product Databases"

_biomolecules, 2022, doi:10.3390/biom12091202_

Round 1

Reviewer 1 Report

This is a well written and informative review on natural product databases and relative chemoinformatics approaches. This manuscript is of great interest in the area of database development.

There are minor corrections/implementations that should be applied:

1) Page 2: "One of the strategies and techniques of drug discovery from NPs is the information based on plants with known activities, including the information either of the therapeutic or the side or toxic effects [10]. Another strategy is the stress-driven growth of plants and microorganisms, which stimulates the production of secondary metabolites [26]." These sentences should be reformulated as information and stress-driven growth are used in drug discovery strategies from NPs but they are not the strategies.

2) Page 3: "The main goal of this manuscript is to review the present state of the art in developing Latin American NP databases and their practical applications to the drug discovery area. Additionally, it discusses the importance of NPs as a source of bioactive molecules, the relevance of compound databases in drug discovery research, and the role of chemoinformatics in the development and analysis of compound databases."

The manuscript reviews first the importance of natural products as a source of bioactive molecules, the relevance of compound databases in drug discovery research, the role of chemoinformatics in the development and analysis of compound databases, the natural product databases and then at the end the Latin American natural product databases. I suggest the authors to keep this order in the above sentence.

3) Page 5: reference 43 should be https://doi.org/10.1021/jm010548w and I suggest the authors to  include also the hit rate from docking in the text for a direct comparison.

4) Page 6, Table 1:  I suggest to include the Chemical Universe Database GDB, https://doi.org/10.1021/ci300415d.

5) Page 8: "Furthermore, with KNIME it is possible to characterize the molecular complexity through the calculation of the number of stereocenters and the fraction of carbon atoms with sp3 hybridization."  This is possible also through the direct use of RDKit.

6) Page 8: KNME --> KNIME

7) Page 12: "Universidade Estadual de Feira de Santana" should be in italic.

8) Page 13: "Unidad de Informática del Instituto de Química" should be in italic.

9) Page 16: "EC50, half maximal inhibitory concentration". Please fix.

Author Response

This is a well written and informative review on natural product databases and relative chemoinformatics approaches. This manuscript is of great interest in the area of database development. There are minor corrections/implementations that should be applied:

RESPONSE: We thank the reviewer for the positive comments to the review manuscript. We did all the suggested modifications. 

1) Page 2: "One of the strategies and techniques of drug discovery from NPs is the information based on plants with known activities, including the information either of the therapeutic or the side or toxic effects [10]. Another strategy is the stress-driven growth of plants and microorganisms, which stimulates the production of secondary metabolites [26]." These sentences should be reformulated as information and stress-driven growth are used in drug discovery strategies from NPs but they are not the strategies.

RE: We modified the sentences to clarify the meaning.

2) Page 3: "The main goal of this manuscript is to review the present state of the art in developing Latin American NP databases and their practical applications to the drug discovery area. Additionally, it discusses the importance of NPs as a source of bioactive molecules, the relevance of compound databases in drug discovery research, and the role of chemoinformatics in the development and analysis of compound databases."

The manuscript reviews first the importance of natural products as a source of bioactive molecules, the relevance of compound databases in drug discovery research, the role of chemoinformatics in the development and analysis of compound databases, the natural product databases and then at the end the Latin American natural product databases. I suggest the authors to keep this order in the above sentence.

RE: We thank the reviewer for this suggestion. The modification was done.

3) Page 5: reference 43 should be https://doi.org/10.1021/jm010548w and I suggest the authors to  include also the hit rate from docking in the text for a direct comparison.

RE: We fixed the reference and included the hit rate from docking. 

4) Page 6, Table 1:  I suggest to include the Chemical Universe Database GDB, https://doi.org/10.1021/ci300415d.

RE: We added GDB to Table 1 and its reference.

5) Page 8: "Furthermore, with KNIME it is possible to characterize the molecular complexity through the calculation of the number of stereocenters and the fraction of carbon atoms with sp3 hybridization."  This is possible also through the direct use of RDKit.

RE: In agreement with the reviewer, we changed “KNIME” to “RDKit”.

6) Page 8: KNME --> KNIME

RE: We did the modification.

7) Page 12: "Universidade Estadual de Feira de Santana" should be in italic.

RE: The name was changed to italics.

8) Page 13: "Unidad de Informática del Instituto de Química" should be in italic.

RE: The name was changed to italics.

9) Page 16: "EC50, half maximal inhibitory concentration". Please fix.

RE: We fixed the definition of EC50 as indicated.

Reviewer 2 Report

Review for the manuscript «Progress and impact of Latin American natural product databases»

The manuscript is well structured and organized, with the main topic being placed into the necessary broader context. To my mind, the paper scientifically soundly addresses the tasks delineated in the introduction. Different significant aspects of the NP databases creation, characterization and utilization are well described with the relevant references for further in detail study. The latter well corresponds to the review format.

I recommend to publish the paper with minor revision, necessary to correct typos and a few awkwardly sounding in English sentences (however, with clear meaning).

I’d suggest to carefully re-read the entire manuscript, correct all typos, messed up references and re-phrase several sentences that are not quite English-looking (e.g. where the subject is missing or following the predicate).

A few examples to focus attention (not a complete list):

1. p.1, “ Captopril was developed based on the bradykinin potentiating factor in Bothrops jararaca snake venom. Is the first animal toxin-based drug approved for human use in 1981 [23,24].”

It seems that “It” is missing in the start of the second sentence.

2. the reference to PADFrag in Table 1 is not correct. Please re-check all the references.

3. p.7, “To say some: ” should be better substituted with “To name a few: “

4. p.7, “Elsewhere [71] have been reviewed the application of this technology…” - subject/predicate mess

5. regarding the terms, “CLogP” seems to have a broader use and context than “SlogP” does

6. p.9, “... it has more than 229,00 Nps.” - one ending 0 is missing (compared to table 2)

To sum up, the overall impression is good and the paper should be published after being polished at the language and typos levels.

Author Response

The manuscript is well structured and organized, with the main topic being placed into the necessary broader context. To my mind, the paper scientifically soundly addresses the tasks delineated in the introduction. Different significant aspects of the NP databases creation, characterization and utilization are well described with the relevant references for further in detail study. The latter well corresponds to the review format.

I recommend to publish the paper with minor revision, necessary to correct typos and a few awkwardly sounding in English sentences (howeveone r, with clear meaning).

RESPONSE: We thank the reviewer for the encouraging and positive comments to the review manuscript. We also thank you for pointing out the sentences with unclear meaning. As advised, we performed a comprehensive review of the style and clarity of the sentences.

I’d suggest to carefully re-read the entire manuscript, correct all typos, messed up references and re-phrase several sentences that are not quite English-looking (e.g. where the subject is missing or following the predicate).

RESPONSE: As advised, we carefully re-read the entire manuscript and fix the issues including, but not limited to the examples pointed out by the reviewer.

A few examples to focus attention (not a complete list):

  1. p.1, “ Captopril was developed based on the bradykinin potentiating factor in Bothrops jararaca snake venom. Is the first animal toxin-based drug approved for human use in 1981 [23,24].”

It seems that “It” is missing in the start of the second sentence.

RESPONSE: “It” was included.

  1. the reference to PADFrag in Table 1 is not correct. Please re-check all the references.

RESPONSE: We corrected the reference to PADFrag and re-checked all the references. 

  1. p.7, “To say some: ” should be better substituted with “To name a few: “

RESPONSE: We did the modification.

  1. p.7, “Elsewhere [71] have been reviewed the application of this technology…” - subject/predicate mess

RESPONSE: The sentence was fixed.

  1. regarding the terms, “CLogP” seems to have a broader use and context than “SlogP” does

RESPONSE: We agree with the comment. For the review paper we changed SLogP to LogP to mention the octanol/water partition coefficient property in general. CLogP and SLogP are two different methods to compute this property.

  1. p.9, “... it has more than 229,00 Nps.” - one ending 0 is missing (compared to table 2)

RESPONSE: We made the modification. Thanks for bringing this typo to our attention.